# Study on the Effect of Judgment Excitation Mode to Relieve Driving Fatigue Based on MF-DFA

**DOI:** 10.3390/brainsci12091199

**Published:** 2022-09-06

**Authors:** Fuwang Wang, Hao Wang, Xin Zhou, Rongrong Fu

**Affiliations:** 1School of Mechanic Engineering, Northeast Electric Power University, Jilin City 132012, China; 2College of Electrical Engineering, Yanshan University, Qinhuangdao 066004, China

**Keywords:** driving fatigue, EEG, hurst exponent, JEM, multifractal spectrum, MF-DFA

## Abstract

Driving fatigue refers to a phenomenon in which a driver’s physiological and psychological functions become unbalanced after a long period of continuous driving, and their driving skills decline objectively. The hidden dangers of driving fatigue to traffic safety should not be underestimated. In this work, we propose a judgment excitation mode (JEM), which adds secondary cognitive tasks to driving behavior through dual-channel human–computer interaction, so as to delay the occurrence of driving fatigue. We used multifractal detrended fluctuation analysis (MF-DFA) to study the dynamic properties of subjects’ EEG, and analyzed the effect of JEM on fatigue retardation by Hurst exponent value and multifractal spectrum width value. The results show that the multifractal properties of the two driving modes (normal driving mode and JEM) are significantly different. The JEM we propose can effectively delay the occurrence of driving fatigue, and has good prospects for future practical applications.

## 1. Introduction

After an extended period of driving, a driver’s attentiveness, feeling, perception, thinking, judgment, will, decision-making capabilites, and control of movement decline to varying degrees, and that driver is shown to enter a state of fatigue. Driving fatigue is one of the leading causes of fatal traffic accidents worldwide [1]. Driving fatigue can lead to decreased driver alertness and poor psychological condition. Long-term driving in this impaired state will damage a driver’s physical and mental health, and is closely related to the occurrence of traffic accidents [2]. When a driver is in a state of mental fatigue, the possibility of unconscious driving arises [3], which seriously endangers the safety of the driver as well as that of the surrounding vehicles and pedestrians. Therefore, accurate fatigue detection and effective fatigue mitigation are particularly important for traffic safety.

Existing driving fatigue detection methods are divided into four categories: detection methods based on subjective factors, detection methods based on vehicle driving characteristics [4,5,6], detection methods based on machine vision characteristics [7,8,9], and detection methods based on human physiological signals [10,11,12,13]. Among them, detection methods based on subjective factors are generally used as auxiliary methods for driving fatigue detection. Detection methods based on vehicle driving characteristics and machine vision characteristics are easily affected by road conditions and light levels. In contrast, detection methods based on human physiological signals have accurate detection and the highest reliability. Commonly used human physiological signals in the field of fatigue driving include electroencephalogram (EEG), electrooculography (EOG), electrocardiogram (ECG), and electromyography (EMG). EEG, which can reflect the characteristics of human brain activity, is one of the most important human physiological signals and is considered to be the most reliable indicator of a fatigued driving state. As a nonlinear dynamical time series [14], EEG is studied using some traditional linear methods that lose many of the inherent properties of a scale-free (self-affine) dynamical signal [15]. For example, the method based on power spectrum is a poor descriptor of the local irregularity of a signal; the method based on Fourier transform can not describe the basic information of neuron dynamics of non-periodic and non-stationary signals; the method based on brain network requires a large number of redundant channels of leads, which have large noise interference and high computational complexity. For efficient handling of nonlinear time series, Mandelbrot put forward fractal theory in 1967 [16], and proposed the multifractal method on this basis [17]. Compared with single fractal, the multifractal method has a multiple scale exponent, which can describe complex time series in more detail through multiple fractal dimensions. In 2002, Kantelhardt combined multifractal and detrended fluctuation analysis methods and proposed MF-DFA [18]. MF-DFA is an effective tool for analyzing nonlinear systems in fractal theory [19] and its fractal characteristics can be studied according to the self-similarity [20] in the waveform details of complex nonlinear dynamic time series. Todd Zorick et al. extracted and analyzed the EEG features of mild explosive traumatic brain injury through a multifractal spectrum, and accurately measured the degree of cognitive and functional damage [21]. Yuan Wz et al. extracted and analyzed the energy parameter characteristics of protein dynamics via Hurst exponent value, and found that the energy, pressure, or volume parameters of protein have obvious fractal characteristics [22]. Jordan Landers and others analyzed the EEG signals of patients with Alzheimer’s disease via multifractal features, and found that MF-DFA is sensitive to the clinical stages of patients and can accurately evaluate the cognitive ability of patients [23]. Jian Wang et al. analyzed the characteristics of brain magnetic resonance imaging (MRI) images via Hurst exponent, and accurately and automatically classified the MRI of normal and abnormal people [24]. Vladimir Matic et al. extracted and analyzed the EEG characteristics of newborns via Hurst exponent value and multifractal spectrum, and effectively distinguished the abnormal EEG activities of sick newborns to different degrees [25]. The above research effectively analyzed the fractal characteristics of human physiological signals via multifractal parameters. In this paper, the multifractal method is applied to the field of driving fatigue, and the fatigue characteristics of EEG signals during driving are analyzed by MF-DFA.

Matteo Chiesi et al. proposed an open source framework called Creamino. It consists of an Arduino-based cost-effective quick-setup EEG platform built with off-the-shelf components and a set of software modules that easily allow users to set up a wide number of neuroscientific experiments [26]. A.P.A. Bueno et al. summarized the existing evidence for eye tracking in neurodegenerative diseases and its potential clinical impact on cognitive assessment. After a series of comparative analyses, they concluded that eye tracking is a promising approach not only for cognitive diagnosis, but more importantly, for tracking underlying cognitive disorders in progressive neurodegenerative diseases [27]. Jozsef Katona et al. proposed an eye-tracking system based on a traditional knowledge level test, analyzing the readability, query, and method syntax of two options in the Language Integrated Query (LINQ) abstraction layer with the same semantics, but different syntax. The results show that the application of eye-tracking systems in the study of complex cognitive processes such as programming is very suitable [28]. Systems such as cost-effect BCI, eye-tracking, etc. have played a very good role in testing cognitive ability and detecting the state of subjects, and have very good prospects for detecting driving fatigue. However, there are few existing fatigue relief methods, and we hope to propose an effective fatigue relief method based on a human-computer interaction system.

Effective fatigue mitigation methods can delay the occurrence of fatigued driving behavior, make drivers maintain a high degree of vigilance, and improve driving performance. Traditional methods to alleviate fatigue include listening to music [29,30,31], using fragrance to stimulate smell [32], intermittent rest [33], and adjusting working time and intensity [34]. However, these methods of relieving fatigue are inefficient because they do not consider the real cause of driving fatigue. Monotonous driving in low-density traffic for a long period of time is an important reason for drivers’ ‘underload’ driving fatigue [35]. In this case, we hope to increase drivers’ active cognitive load and improve driving performance by introducing appropriate secondary cognitive tasks and encouraging drivers.

In this study, we propose JEM, which can improve a subjects’ driving interest and increase the active cognitive load through the human–computer interaction of voice + button, thereby achieving the goal of delaying the occurrence of fatigue. We studied the dynamic characteristics of subjects’ EEG via MF-DFA, and analyzed the fatigue-delaying effect of JEM via Hurst exponent and multifractal spectrum width. The results showed that during the continuous human–computer interaction, the activity of EEG was greatly improved, and the development speed of fatigue was slowed.

## 2. Materials and Methods

### 2.1. Subjects

We recruited volunteers and randomly selected 12 subjects (10 males and 2 females, the ratio of males to females in road traffic accidents in China is 17:3 [36], so the total sample size of 12 should include 10 males and 2 females; age 24 ± 1.6 (standard deviation, SD)) who met the requirements for the experiment. All the subjects were healthy and had driving licenses. They did not drink any irritant drinks, eat any irritant foods, or take any irritant drugs within the 48 h preceding the experiment. Before the experiment, the subjects were informed of the research content and experimental process, and signed the consent form. The ethics committee of the hospital of northeast electric power university approved the research scheme according to the ethics code of the World Medical Association (Helsinki Declaration).

### 2.2. Procedure and EEG Recording

In this experiment, a JT/T378 automobile simulator is used to simulate driving. This simulator has an interactive visual system, and is able to record and prompt incorrect driving operations (gear mismatch, collision, speeding, red light running, etc.). The relative positions of the control parts are consistent with those of a real car. The strength is also consistent with that of a real car, the performance of the operating parts is reliable, the operation is flexible, and the noise is low. In order to ensure the authenticity of the experiment, we selected an urban road scene with moderate traffic flow, wide roads, and good visibility for the experiment. The scene includes straight lanes, left turns, right turns, intersections, and daily traffic signs, and required the subjects to obey the traffic rules while driving.

In this experiment, a Neuroscan is used to collect EEG data. Its electrodes are connected to the scalp according to the international 10–20 system (30 channels =FP1, FP2, F7, F3, FZ, F4, F8, FT7, FC3, FCZ, FC4, FT8, T3, C3, CZ, C4, T4, TP7, CP3, CPZ, CP4, TP8, T5, P3, PZ, P4, T6, O1, Oz, and O2). The Neuroscan can accurately detect the characteristics of all brain regions. Previous studies have shown that the features of brain fatigue can be detected in the EEG signals of the central region of the human brain [37], so this study uses the data collected by the central region (C3) channel.

This experiment is divided into normal driving mode and JEM. Each subject completes normal driving mode on the first day and JEM on the second day. All subjects drove continuously for 3 h from 1:00 p.m. to 4:00 p.m. In order to avoid the impact of lack of sleep on the experimental results, the subjects were arranged to rest for half an hour (12:00 p.m–12:30 p.m.) before the beginning of the experiment. The data acquisition process is divided into four stages: first stage (1:00 p.m–1:05 p.m.), second stage (2:00 p.m–2:05 p.m.), third stage (3:00 p.m–3:05 p.m.), and fourth stage (3:55 p.m–4:00 p.m.). The subjects were asked about their current status five minutes before each stage, and the drivers were scored according to Borg’s category rating scale (CR 10-scale). In the process of data collection, subjects stopped driving, sat still and looked ahead, avoiding redundant actions to ensure the accuracy of the collected data.

The hardware for JEM mainly includes the response switch, microcontroller (MCU), speech synthesis module (SYN6288), speech chip (ISD4004), and speaker. For MCU, the model we selected was the Arduino UNO R3.

Figure 1 shows the experimental environment and the experimental equipment used.

### 2.3. JEM

The effects of increasing cognitive load have previously been studied. Point Gershon et al. kept the subjects on high alert by constantly asking the subjects questions while driving [38]. Maria Evstigneeva et al. added active cognitive tasks and passive cognitive tasks to the subjects’ driving process, respectively, and found that the fatigue improvement effect of cognitive tasks with active discrimination was better than that of cognitive tasks with passive discrimination [39]. Taloron Gilad et al. evaluated the effect of vigilance maintenance tasks during driving through comparative experiments, and found that trivia tasks would decrease subjects’ fatigue and improve driving performance, while working memory tasks and selective response time tasks would increase the subjects’ subjective fatigue [40]. Leila Takayama et al. added passive media and slightly interactive media, respectively, during subjects’ driving process, and found that slightly interactive media caused better fatigue relief effect than passive media [41]. Yu Kai Wang et al. added visual stimulation to the subjects’ driving, and the results showed that visual stimuli increased the subjects’ reaction time, and the effect of delaying fatigue was weaker than that of voice stimuli [42]. Zhaohui Huang et al. studied different human–computer interaction modes in driving, and believed that the mode with visual interaction was not suitable for driving behavior, while the dual channel human–computer interaction mode similar to voice gesture was more suitable for driving [43].

The cognitive task of this experiment adopts the JEM. First, before the experiment, we stored the subjects’ favorite athletes, actors, singers, and familiar friends and their related information in the database by means of questionnaires to ensure that the subsequent judgment information could arouse the interest of the subjects. Then, we collected and sorted out enough data to clearly describe the above people, and set them as problems for judgment information. Finally, we collected and sorted out excitation information. Since we conducted a comprehensive analysis of the results of the subjects’ questionnaires, the excitation information for different subjects was also completely in line with the subjects’ interests, so we could ensure that the excitation information would strongly arouse the subjects’ interests. The excitation information corresponding to the athlete provided an exciting instant explanation of the athlete winning the championship; the excitation information corresponding to the actor was the actor’s impassioned or moving lines; the excitation information corresponding to the singer was the singer’s beautiful song; the excitation information corresponding to a familiar friend was the voice information of the friend recorded in advance.

In the JEM, the MCU controls the speech synthesis module to play the judgment information through the loudspeaker every five minutes. The subjects need to judge the described person according to the judgment information and repeat the judged keywords orally, then select through the left and right keys equipped on the steering wheel. If the subjects choose correctly, MCU controls the voice chip to play the excitation information corresponding to the described person through the speaker; if the subject makes an incorrect selection, the speaker will play an error tone for 3 s.

Yi Ching Lee et al. believe that inappropriate cognitive tasks will deepen driver fatigue [44]. Therefore, we added the emergency stop test in the experiment to ensure that our JEM would not affect the subjects’ ability to deal with unexpected problems. At 1:45 p.m., 2:45 p.m. and 3:45 p.m. in the two driving modes (normal driving mode and JEM), the word ‘stop’ was played through the speaker, and the subjects were asked to step on the brake immediately after hearing the word. Their reaction times were recorded.

### 2.4. Methods

EEG is the potential change produced by nerve cells in the cerebral cortex, which contains rich physiological information. Previous studies used *θ* (4~7 Hz) and *β* (14~32 Hz) sub-bands to effectively detect the driver’s fatigue status. Among them, the *θ* sub-band will increase significantly when the driver is tired, while the *β* sub-band will decrease significantly [12].

In this experiment, the extracted *θ* and *β* sub-bands were used to obtain and compare the fatigue characteristics of the two driving modes through the MF-DFA, so as to verify the mitigation effect of our proposed JEM on driving fatigue.

#### 2.4.1. MF-DFA

MF-DFA, proposed by Kantelhardt et al., is applicable to the analysis of a non-stationary time series [15]. This method can generally be described as: There is a non-stationary time series {*x_k_*} (*k* = 1, 2, 3… *n*). We will get the corresponding random walk time series and calculate the fractal exponent. The steps are as follows:

Calculate the average value of the whole time series. As shown in Equation (1):(1)$$\bar{x}=\frac{1}{n}\sum_{k}^{n}x_{k}$$

Subtract the mean calculated by Equation (1) from the time series. As shown in Equation (2):(2)$$Y(i)=\sum_{k=1}^{i}\left[x_{k}-\bar{x}\right]$$

By Equation (2), {*Y*(*i*)} is the required random walk time series. The series was then further analyzed using multifractal methods.

{*Y*(*i*)} is divided into non-overlapping intervals *v*, the number of intervals is *N*, and the length is *s* (*s* is called the time scale). However, when *s* cannot completely divide the random walk time series {*Y*(*i*)}, data loss will occur. If {*Y*(*i*)} is divided again from the opposite direction with length *s*, 2*N* time series that do not overlap one another can be obtained, this method avoids the occurrence of missing data [45]. 

All intervals are fitted by the least square method to obtain the fitting trend value. Its definition is shown in Equation (3):(3)$$y_{v}(i)=a_{1}i^{k}+a_{2}i^{k-1}+\ldots a_{k}i+a_{k+1}$$
where, i=1,2,3,…,s,k=1,2,…,2N.

The local trend of the interval can be expressed by fitting residual, and the calculation method has the values of all points on each interval *v* be subtracted from the fitting trend value *y_v_*(*i*). The scale function *F*^2^(*v*,*s*) can be expressed by the mean square value of the local trend. 

When *v* = 1,2, …, *N*, *F*^2^(*v*,*s*) is shown in Equation (4):(4)$$F^{2}(v,s)=\frac{1}{s}\sum_{i=1}^{s}{\left\{Y[(v-1)s+i]-y_{v}(i)\right\}}^{2}$$

When *v* = *N* + 1, *N* + 2, …, 2*N*, *F*^2^(*v*,*s*) is shown in Equation (5):(5)$$F^{2}(v,s)=\frac{1}{s}\sum_{i=1}^{s}{\left\{Y[n-(v-N)s+i]-y_{v}(i)\right\}}^{2}$$

The number of points contained in the interval will affect the scale function. When there are fewer data points, the fast wave will affect the scale function, while when the data points are longer, the slow wave will affect the scale function. Therefore, in order to highlight the influence of fast waves and slow waves on the time series, the time series is divided into multiple segments of different sizes in the calculation of the scale function, denoted asRMS{ns}(v), as shown in Equation (6):(6)RMS{ns}(v)=F(v,s)
where, *n_s_* represents the number of different time periods, and *v* represents an interval.

The *q*th-order fluctuation function is obtained by calculating the average value of all intervals. When *q* satisfies the condition of *q* ≠ 0, *F_q_*(*s*) is expressed as Equation (7):(7)$$F_{q}(s)={\left\{\frac{1}{2N}{\sum_{v=1}^{2N}\left[F^{2}(v,s)\right]}^{\frac{q}{2}}\right\}}^{\frac{1}{q}}$$the variable *q* in Equation (7) can take any value except 0. For example, when *q* = 2, it is equivalent to the standard DFA program, and the fluctuation function increases with the increase of *s*. When the value of *q* = 0, Equation (7) is transformed into Equation (8):(8)$$\ln[F_{q}(s)]=\frac{1}{2N}\sum_{v=1}^{2N}\ln[F_{v}(s)]$$

When it is determined that {*x_k_*} is a long-range correlation time series, there is a relationship of Equation (9):(9)$$F_{q}(s)\propto s^{h(q)}$$

Equation (9) shows that the fluctuation function has an exponential relationship with the number of points in the divided interval, and logarithms are taken from both sides, as shown in Equation (10):(10)$$\mathrm{lg}F_{q}(s)=h(q)\mathrm{lg}s+C$$
(11)$$h(q)=\frac{\log F_{q}(s)}{\log s}$$

According to Equation (11), *h*(*q*) (Hurst exponent) is the slope obtained by linear regression. Hurst exponent was proposed by H.E Hurst of Britain. It is used to analyze the time correlation of time series and to judge whether there is chaos in the data, and is sensitive to slight changes in the characteristics of the series [46]. The scaling behavior with different amplitude fluctuations can be described according to the *q* value. For example, the scaling behavior with large amplitude fluctuation corresponds to a larger *q*, and the scaling behavior with small amplitude fluctuation corresponds to a smaller *q*. A single order corresponds to a Hurst exponent, which depends on *q* and is called *q*th-order Hurst exponent, as shown in Equation (11). For example, when *q* = 2, the scale exponent *h*(2) is called the classical Hurst exponent.

Transform *q*th-order *h*(*q*) into *q*th-order mass exponent τ(*q*), as shown in Equation (12):(12)τ(q)=qh(q)−1

Equation (12) explains the relationship between Hurst exponent and mass exponent. As a nonlinear function, the mass exponent is often used to calculate the singularity exponent and singularity dimension. If the mass exponent exhibits linear characteristics, it means that the time series is a single fractal process. 

The size of the singularity exponent *α* represents the different degrees of singularity of the interval. The fractal singularity dimension *f*(*α*) describes the fractal dimension based on the singularity strength and represents the distribution density of the corresponding singularity exponent. 

The singularity exponent *α* and *q*th-order singularity dimension *f*(*α*) are calculated by Equations (13) and (14):(13)α=τ’(q)
(14)f(α)=qα−τ(q)
in which, *α* represents the different degrees of singularity corresponding to each interval, and *f*(*α*) represents the fractal dimension with the singularity exponent *α*. The *α-f*(*α*) curve is generally called multifractal spectral, and its shape is single-peak arch.

Equation (15) represents the width values of the fractal spectrum:(15)$$\Delta \alpha =\alpha_{\max}-\alpha_{\min}$$

The width values not only reflects the inhomogeneity and complexity of the time series, but also reflects the strength of the brain’s ability to process relevant information and the strength of neural activity. Larger values of singular spectral width correspond to more complex distributions and larger fractal strengths [47].

#### 2.4.2. Statistical Analysis Algorithm

The statistical analysis method of this experiment adopts the two tailed *t*-test, which can compare whether there is a significant difference between the two data samples. In the next study, we will analyze the significant difference of driving fatigue by using the two tailed *t*-test to analyze the MF-DFA attribute eigenvalues of the two driving modes.

## 3. Results

### 3.1. Subjective Questionnaire

Previous studies have shown that a subjective questionnaire is a better method to assist in detecting fatigue [48]. In this experiment, Borg’s category rating scale (CR 10 scale) (0 = nothing at all, 0.5 = extremely weak, 1 = very weak, 2 = weak, 3 = moderate, 5 = strong, 7 = very strong, 10 = extremely strong) is used to evaluate the different fatigue states of drivers. The results are shown in Figure 2.

In Figure 2, the fatigue degree of subjects in both driving modes showed an increasing trend with the progression of driving stages. However, it is obvious that the speed of subjects’ fatigue increase in the JEM is lower than that in the normal driving mode, and there is a significant difference (|t| = 2.946, t_0.05,12_ = 2.179, *p* < 0.05) between the two driving modes. This means that the JEM can improve the subjects’ driving interest and neural activity by increasing cognitive load. The results of the subjective questionnaire show that the proposed JEM can effectively alleviate subjects’ driving fatigue.

### 3.2. Multifractal Detrended Fluctuation Analysis

#### 3.2.1. Hurst Exponent

We calculate the Hurst exponent through Equation (11), and Figure 3 shows the Hurst exponent of *θ* sub-band for a subject in the two driving modes.

Previous studies have shown that Hurst exponent describes the scaling and power law of signals. For time series signals, if the Hurst exponent shows a regular change trend with the change of *q*, the sequence has obvious multi-scale characteristics [49]. In Figure 3, *H(q)* is a nonlinear function of *q*, which gradually decreases with the increase of *q*. It shows the multi-scale characteristics of the time series signal. For different *q*, there are different exponential powers, representing different power-law autocorrelations. The regular change of Hurst exponent range corresponding to different driving stages also proves the existence of long-range correlation. The Hurst exponent of the *θ* sub-band in both driving modes in Figure 3a,b increases gradually with the duration of driving time, indicating that during the driving time from the first stage to the fourth stage, the volatility and complexity of the *θ* sub-band gradually increases. Since the *θ* sub-band usually appears with fatigue, indicating that the subject has gradually changed from an awake state to a fatigued state, the Hurst exponent clearly shows this fatigue trend.

Figure 4 shows the comparison of the Hurst exponents of the *θ* sub-band for a subject corresponding to the two driving modes under the four driving stages. In Figure 4, we can see that when the subject is in the first stage (Figure 4a), the Hurst exponent width values are basically the same and the curves overlap, but in the second (Figure 4b), third (Figure 4c), and fourth (Figure 4d) stages, the Hurst exponent width value in the JEM mode is smaller than that in the normal driving mode. The Hurst exponents for the two driving modes were significantly different (|t| = 3.293, t_0.05,12_ = 2.179, *p* < 0.05). Since the subject had just entered the driving state when the first stage data was collected, the subject was awake in both modes at this time. As the driving time continued, the fatigue degree of the subject gradually deepened. However, due to the continuous stimulation of the subject through judgment and excitation in the JEM, the cognitive load and driving interest of the subject was increased, and the occurrence of fatigue in the subject was effectively delayed. Therefore, in the JEM, the growth rate of the volatility and complexity of the *θ* sub-band accompanying the fatigue state is relatively slow, which is significantly smaller than that of the normal driving mode.

Figure 5 shows the Hurst exponent of the *β* sub-band for a subject in the two driving modes. In Figure 5a,b, the Hurst exponent of the *β* sub-band in two driving modes gradually decreases with continued driving time, indicating that the volatility and complexity of the *β* sub-band gradually decreases during the driving time from the first stage to the fourth stage. Since the *β* sub-band usually appears in the awake state, the change in the Hurst exponent clearly describes the whole process of the subject’s gradual transition from the awake state to the fatigue state.

Figure 6 shows the comparison of the Hurst exponent of the *β* sub-band for a subject corresponding to the two driving modes under the four driving stages. According to Figure 6, we can clearly see the obvious difference in Hurst exponents between the two driving modes. When the subject is in the first stage (Figure 6a), the Hurst exponent width values are basically the same and the curves coincide, but in the second (Figure 6b), third (Figure 6c), and fourth (Figure 6d) stages, the Hurst exponent width value in JEM mode is greater than that in normal driving mode, and there is a significant difference between the Hurst exponent values in the two driving modes (|t| = 2.848, t_0.05,12_ = 2.179, *p* < 0.05). Since the subject had just entered the driving state when the first stage data was collected, the subject was awake in both modes at this time. As the driving time continued, the degree of the subject’s fatigue gradually deepened. However, due to the continuous stimulation of the subject through judgment and excitation in the JEM, the cognitive load and driving interest of the subject was increased, and the occurrence of fatigue of the subject was effectively delayed. Therefore, under JEM, the volatility and complexity of the *β* sub-band that accompanies the awake state are significantly higher than those in the normal driving mode during driving, indicating that the JEM mode plays a good role in reducing fatigue.

#### 3.2.2. Multifractal Spectrum

We calculate the multifractal spectra of the *θ* sub-band of the two driving modes by Equations (11) and (12), and calculate the multifractal spectrum width values by Equation (13). Figure 7 shows the multifractal spectra of the *θ* sub-band of the two driving modes.

In Figure 7, *α* is the q-order singularity exponent, which is the embodiment of the singularity in different intervals. *f(a)* represents the singular dimension under the corresponding singular exponent, also known as the fractal dimension. The width of the multifractal spectrum is recorded as the difference between the maximum and minimum values of *α*, and is widely used as a parameter to measure the strength of the multifractal. Studies have found that multifractal spectra with larger widths tend to have greater fractal intensity than those with smaller widths, and their inhomogeneity and complexity will also be higher [47]. In Figure 7a,b, the multifractal spectral width values of the *θ* sub-band in both driving modes gradually increase with the duration of driving time, indicating the process of driving time from the first stage to the fourth stage, the volatility and complexity of the *θ* sub-band gradually increase. Since the *θ* sub-band usually appears with fatigue, indicating that subject gradually changed from an awake state to a fatigued state, the multifractal spectrum clearly showed this fatigue trend.

Figure 8 shows the comparison of the multifractal spectrums of *θ* sub-bands for a subject corresponding to the two driving modes under the four driving stages. From Figure 8, we can see that the multifractal spectrum curves are basically the same, and overlap when the subject is in the first stage (Figure 8a); however, in the second (Figure 8b), third (Figure 8c), and fourth (Figure 8d) stages, the multifractal spectrum width value in JEM mode is smaller than that in the normal driving mode, and there is a significant difference in the multifractal spectrum width value between the two driving modes (|t| = 2.748, t_0.05,12_ = 2.179, *p* < 0.05). Since the first stage of data collection was at the initial stage of driving, subjects were all awake at this time. As the driving time continued, subjects’ fatigue gradually accumulated. However, JEM improved subjects’ cognitive load and driving interest through judgment and excitation, and effectively delayed subjects’ fatigue. Therefore, in JEM, the volatility and complexity of the *θ* sub-band accompanying fatigue states grow significantly less than in normal driving mode.

Figure 9 shows the multifractal spectrum of the *β* sub-band for a subject in the two driving modes. In Figure 5a,b, the multifractal spectrums of the *β* sub-band in two driving modes gradually decrease with continued driving time, indicating that the volatility and complexity of the *β* sub-band gradually decrease during the driving time from the first stage to the fourth stage. Since the *β* sub-band is usually present in the awake state, the multifractal spectrum clearly shows subject’s gradual transition from awake state to fatigue state as driving progresses.

Figure 10 shows the comparison of the multifractal spectrums of the *β* sub-band for a subject corresponding to the two driving modes under the four driving stages. We can see from the Figure 10 that when the subject is in the first stage (Figure 10a), the multifractal spectrum width values are basically the same, and the curves coincide. However, in the second (Figure 10b), third (Figure 10c), and fourth (Figure 10d) stages, the multifractal spectrum width value in JEM mode is greater than that in normal driving mode, and there is a significant difference between the multifractal spectrum values in the two driving modes (|t| = 2.9430, t_0.05,12_ = 2.179, *p* < 0.05). Since the subjects had just entered the driving state when the first phase of data was collected, the subjects had high neural activity at this time and showed awake states in both modes. With the accumulation of fatigue in the driving behavior, the subjects’ fatigue level gradually deepened. However, due to the continuous stimulation of the subjects through judgment and excitation in JEM, the cognitive load of the subjects were significantly increased, and the generation rate of fatigue was reduced. Thus, the volatility and complexity of the *β* sub-band that accompanies the awake state maintains a low rate of decline under JEM. However, there is no human intervention in the normal driving mode, so the volatility and complexity of the the *β* sub-band decreased faster.

The average values of Hurst exponent range values and multifractal spectrum width values for 12 subjects at different driving stages are shown in Figure 11 and Figure 12.

Through the comparison of the two groups of multifractal features in Figure 11 and Figure 12, and the statistical analysis of the strength of the multifractal attributes in the two driving modes, it is easier to distinguish the different fatigue states of the subjects. For the Hurst exponents and multifractal spectrograms of the *θ* and *β* sub-bands under different driving stages, the multifractal properties of the JEM are significantly better than those in the normal driving mode (Hurst exponent range of *θ* sub-band: |t| = 5.1552, t_0.05,12_ = 2.179, *p* < 0.05; Hurst exponent range of *β* sub-band: |t| = 3.2475, t_0.05,12_ = 2.179, *p* < 0.05; multifractal spectrum width values of *θ* sub-band: |t| = 3.1129, t_0.05,12_ = 2.179, *p* < 0.05; multifractal spectrum width values of *β* sub-band: |t| = 3.3335, t_0.05,12_ = 2.179, *p* < 0.05).

In conclusion, the fractal properties of the subjects’ EEG signals were significantly different in the two driving modes. It can be confirmed that the JEM proposed in this paper has a good fatigue-relieving effect.

### 3.3. Emergency Stop Test

To determine whether our proposed JEM would cause additional cognitive load to the subjects’ normal driving, we performed an emergency stop test, and the results are shown in Figure 13.

In Figure 13, the three-time average parking reaction time of the 12 subjects in the JEM are significantly smaller than that in the normal driving mode (|t| = 6.2476, t_0.05,12_ = 2.179, *p* < 0.05), indicating that our proposed method does not impose additional cognitive load on the subjects’ normal driving behavior, while it does increase the vigilance of the subjects for emergencies.

## 4. Discussion

Previous studies have shown that prolonged driving on monotonous roads can cause ‘underload’ driving fatigue [38]. To alleviate such fatigue and thereby reduce the incidence of traffic accidents [50], we propose JEM. It can effectively delay the occurrence of fatigue by constantly judging and exciting subjects. Meanwhile, as the ‘gold standard’ for fatigue detection [51], EEG exhibits significant nonlinear and dynamic properties [52]. MF-DFA, a recently proposed nonlinear dynamics method, can effectively evaluate multifractal scales of non-stationary data [23]. Therefore, this paper uses MF-DFA to extract, analyze, and compare the fatigue features of the two driving modes.

### 4.1. Previous Studies and This Study

The traditional methods for delaying driving fatigue are inefficient and unsuitable for online real-time relief of driving fatigue, and even have side effects. For example, although the method based on electrical stimulation [53] can relieve the driver’s fatigue to a certain extent, the tolerance level is not the same for different drivers and different stimulated parts [54], and the use of this method will have certain side effects on the human body. The fatigue relief effect of active cognitive tasks with a motor component is better than that of passive cognitive tasks without any motor components [39], but driving is a behavior that requires attention, thus adding motor components to cognitive tasks (such as stretching or straightening the legs to relax them) is a very dangerous proposition. Listening to music during driving is the choice of many drivers at present, and many studies have shown that music has a fatigue-alleviating effect [29,30,31]. However, this method is too monotonous under long-term driving, and the fatigue relief effect of single music stimulation will gradually weaken, resulting in a reduced ability of the driver to maintain interest, thereby entering the fatigue state. When feeling sleepy while driving, some drivers may choose to stop and take a nap to relieve fatigue. Sleep is indeed effective in reducing fatigue, but sometimes drivers cannot stop and rest for various reasons, and due to sleep inertia (the transition from sleep to wakefulness can take up to 30 min), it can be dangerous to drive directly after waking up [55]. Appropriate scents can have certain positive effects on the human body, such as the scent of sweet orange to reduce fatigue, and the scent of mint to ease tension and help maintain vitality. However, previous experiments have shown that olfactory stimuli alone have very limited effects on the subjective experience of subjects [56], and that the effect of such passive stimuli is much lower than that of active cognitive tasks when the driver is driving for a long time [39]. Choice response tasks and working memory tasks have been used frequently in past studies [57,58,59,60], but these two cognitive tasks are not suitable for use in driving fatigue relief. Research has shown that choosing a reaction time task not only did not improve driving performance, it also increased feelings of drowsiness, while working memory tasks actually worsened driving conditions [40]. In our sensory system, vision often dominates [61]. But the introduction of visual stimuli during driving interferes with the driver’s driving behavior [62] and adds additional reaction time [42].

Considering the limitations of the above traditional fatigue mitigation methods, we propose JEM. First, we collected information of interest to the subjects through the questionnaire, and organized it into judgment questions and excitation information to ensure that the subjects had enough attention and good feedback for our information. A performer can improve their performance over a relatively long period of time when motivated by an interest [38]. Second, JEM is a trivial task, and enables the subjects to actively think and judge, which effectively delays fatigue [39,40]. We asked subjects to verbally say keywords that gave an answer when giving the judgment results, and the effect of this slightly interactive media was better than that of passive interactive media [41]. The judgment questions we posed were familiar to the subjects, and they could be answered with just a little thought, so that they did not take up too much of the subjects’ attention, and had no side effects that would interfere with their main driving tasks. In the experiment, the system transmitted information to the subjects through voice, and the subjects could simply feed back the answer to the system through the buttons on the left and right sides of the steering wheel. In the process of task control, the interactive advantages of the two channels of voice + button were fully utilized to successfully complete the JEM. Through this method, increasing the subject’s cognitive load within an acceptable range could significantly improve the subject’s driving performance and relieve fatigue [12]. The experimental device is simple and portable, easy to operate in practical applications, and contributes to improving road safety.

In order to more intuitively show that JEM can reduce accidents and increase road safety by relieving fatigue, we used the previous experiment of music alleviating fatigue in the same driving environment as a reference, which refers to the research results of Rui Li et al. [30]. They concluded that moderate-paced music contributed to long-term improvements in fatigue and concentration levels; compared to the no-music group, slow-paced music only improved attention quality for a short period of time, significantly reducing fatigue and concentration levels; fast-paced music helped relieve long-term driver fatigue, but further worsened driver concentration compared to no music. The music we used in the previous music fatigue alleviation experiment was the moderate-paced music that had shown the best effect on improving fatigue during long-term driving, and the music fatigue alleviation was compared with the JEM fatigue alleviation proposed in this experiment. Table 1 shows the average value of the subjects’ wrong driving operations in the experiment, and the subjects’ wrong driving operations value were recorded and saved by the driving simulator in the previous experiment.

As shown in Table 1, the average number of wrong driving operations of subjects under music to relieve fatigue is higher than the average number of wrong driving operations of subjects under JEM, indicating that our proposed JEM is more effective in relieving fatigue during long-term driving.

Compared with traditional linear and time-frequency methods, the MF-DFA we use is also more suitable for dealing with nonlinear time series such as EEG. For the nonlinear and dynamic properties exhibited by EEG [52], conventional methods ignore many properties inherent in these signals [15]. MF-DFA can analyze non-stationary, complex and variable time series [63], accurately identify the fractal features of biological and physiological systems, including the brain, and fully display the local and different levels of characteristics of the signal through multiple fractal dimensions and local singular values.

### 4.2. Limitations and Future Research Lines

In this study, the JEM we proposed conducts human–computer interaction through two channels of voice + button. Compared with the normal driving mode, the JEM can effectively delay the occurrence of fatigue in subjects. This method is simple and effective in practical applications, but there is no in-depth study on the effect of different types and degrees of difficulty of judgment problems on delaying fatigue, and the intelligence needs to be improved.

In future research, we will subdivide the type and difficulty of the selection of judgment questions to determine the effectiveness of different judgment questions in delaying fatigue. At the same time, we will add voice recognition and artificial intelligence methods on the basis of JEM, so that the entire fatigue relief process can be carried out through a pleasant human–machine dialogue.

## 5. Conclusions

This paper proposes JEM that can delay driving fatigue. Using MF-DFA to compare and analyze the fractal characteristics of EEG of subjects in JEM and normal driving modes, it is found that in the continuous human–computer interaction, the activity of EEG has been greatly improved, and the development speed of fatigue has been alleviated. At the same time, the device for JEM is simple and portable, with low difficulty of operation, and has a good prospects for future practical applications.

## Figures and Tables

**Figure 1 brainsci-12-01199-f001:**
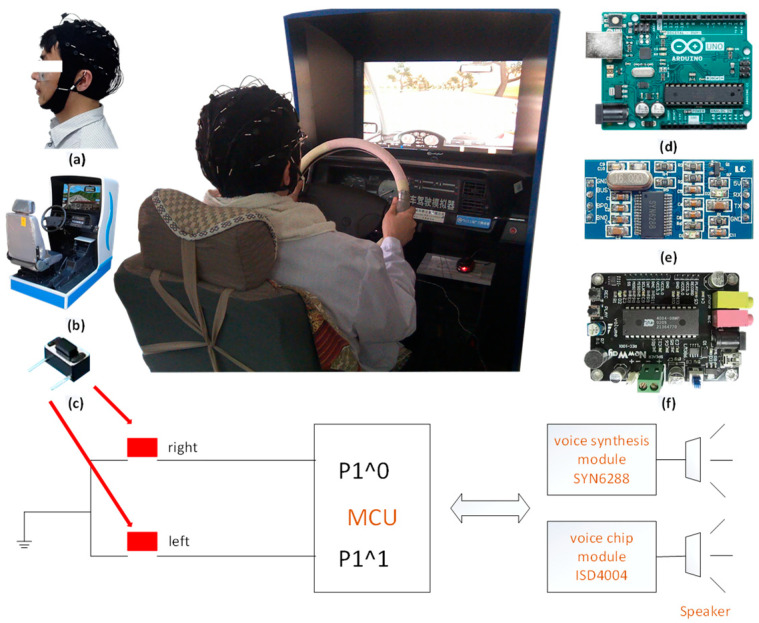
Experimental equipment diagram (**a**) Neuroscan (**b**) JT/T378 (**c**) response switch (**d**) Arduino UNO R3 (**e**) speech synthesis module SYN6288 (**f**) speech chip ISD4004.

**Figure 2 brainsci-12-01199-f002:**
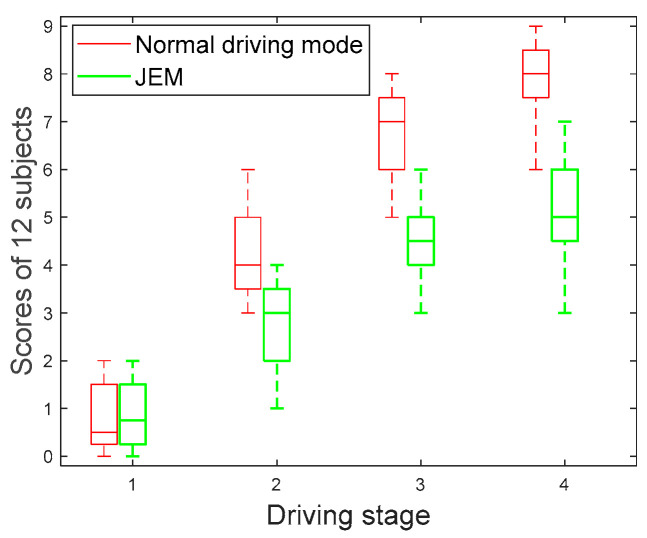
Change trend of subjective questionnaire scores.

**Figure 3 brainsci-12-01199-f003:**
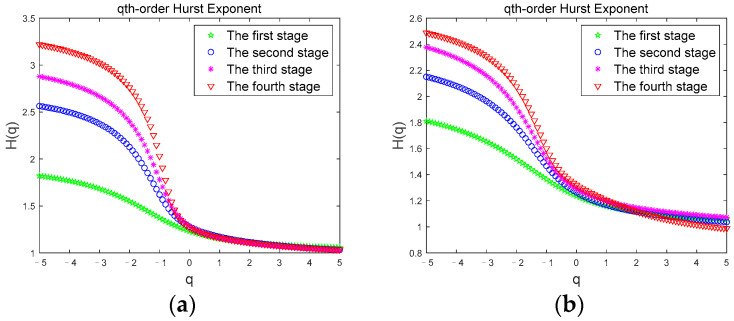
Hurst exponent. (**a**) Hurst exponent of the *θ* sub-band in Normal driving mode; (**b**) Hurst exponent of the *θ* sub-band in JEM.

**Figure 4 brainsci-12-01199-f004:**
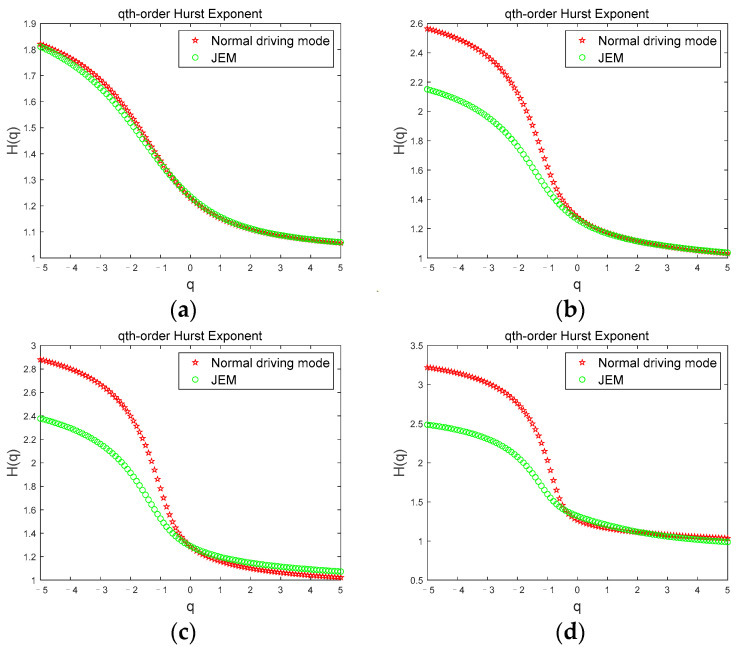
Comparison of the Hurst exponent of the *θ* sub-band in two driving modes. (**a**) first stage; (**b**) second stage; (**c**) third stage; (**d**) fourth stage.

**Figure 5 brainsci-12-01199-f005:**
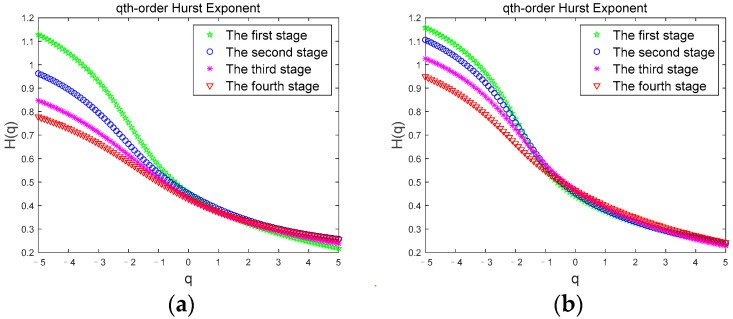
Hurst exponent. (**a**) Hurst exponent of the *β* sub-band in Normal driving mode; (**b**) Hurst exponent of the *β* sub-band in JEM.

**Figure 6 brainsci-12-01199-f006:**
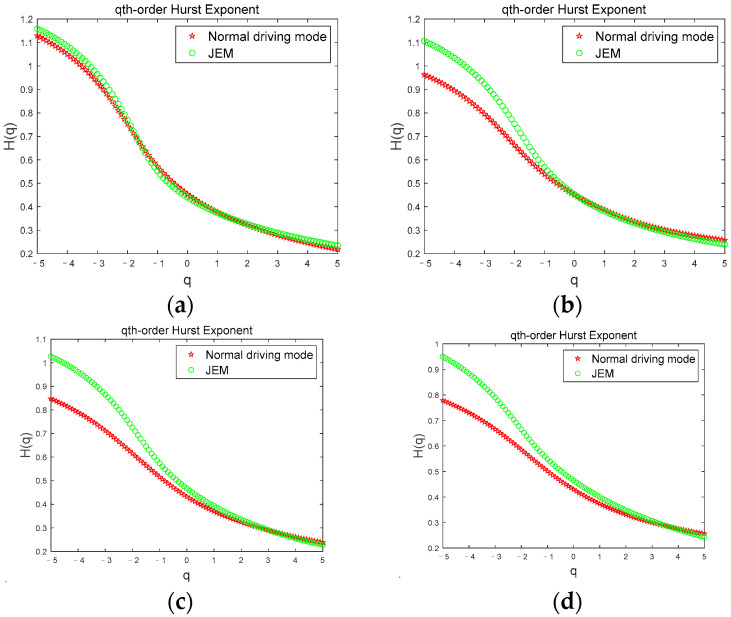
Comparison of the Hurst exponent of the *β* sub-band in two driving modes. (**a**) first stage; (**b**) second stage; (**c**) third stage; (**d**) fourth stage.

**Figure 7 brainsci-12-01199-f007:**
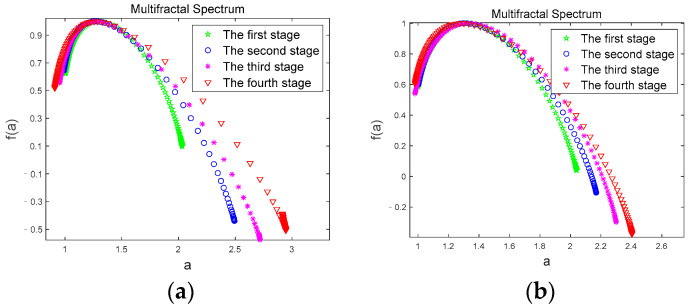
Multifractal spectrum. (**a**) multifractal spectrum of the *θ* sub-band in Normal driving mode; (**b**) multifractal spectrum of the *θ* sub-band in JEM.

**Figure 8 brainsci-12-01199-f008:**
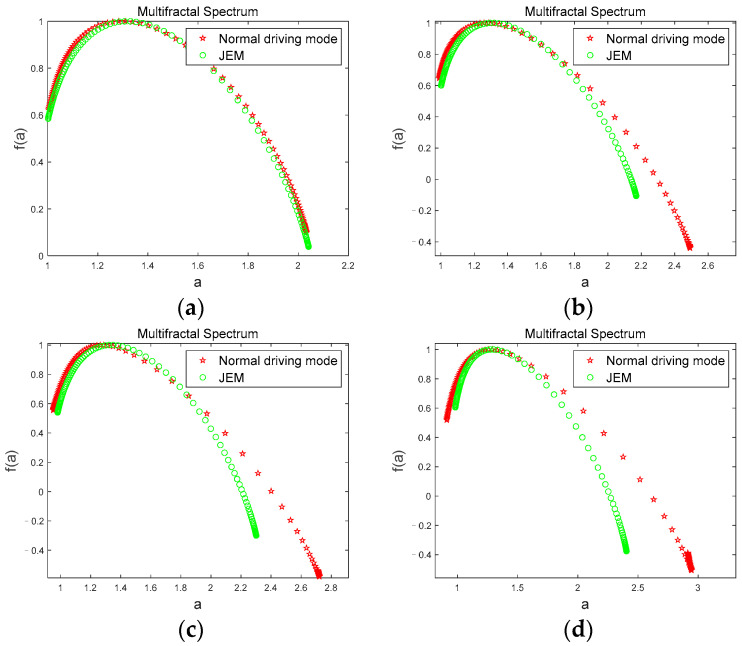
Comparison of multifractal spectrum of the *θ* sub-band in two driving modes. (**a**) first stage; (**b**) second stage; (**c**) third stage; (**d**) fourth stage.

**Figure 9 brainsci-12-01199-f009:**
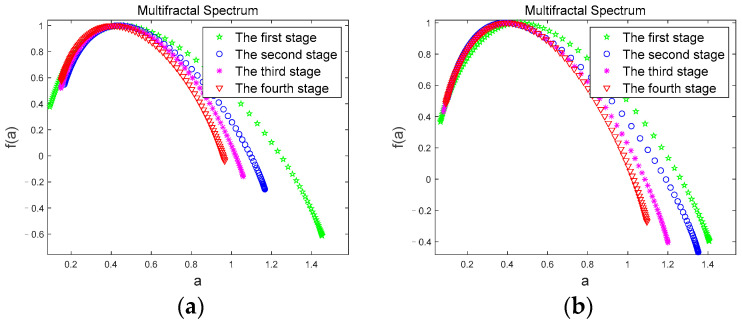
Multifractal spectrum. (**a**) multifractal spectrum of the *β* sub-band in Normal driving mode; (**b**) multifractal spectrum of the *β* sub-band in JEM.

**Figure 10 brainsci-12-01199-f010:**
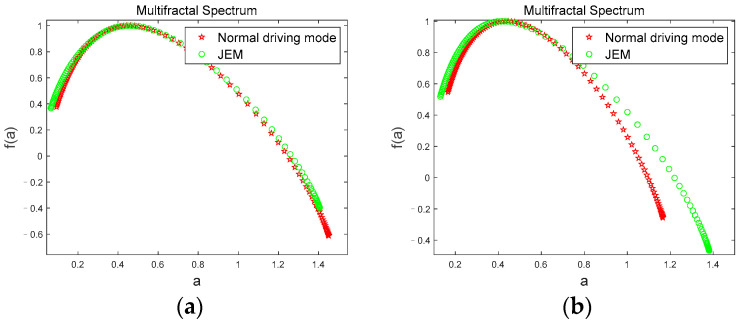
Comparison of the multifractal spectrums of the *β* sub-band in two driving modes. (**a**) first stage; (**b**) second stage; (**c**) third stage; (**d**) fourth stage.

**Figure 11 brainsci-12-01199-f011:**
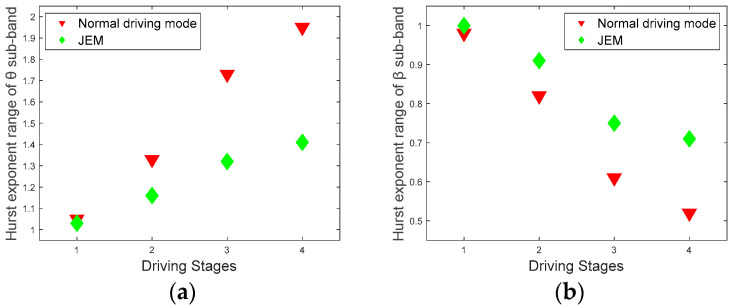
Hurst exponent range. (**a**) Hurst exponent range of the *θ* sub-band in two driving modes; (**b**) Hurst exponent range of the *β* sub-band in two driving modes.

**Figure 12 brainsci-12-01199-f012:**
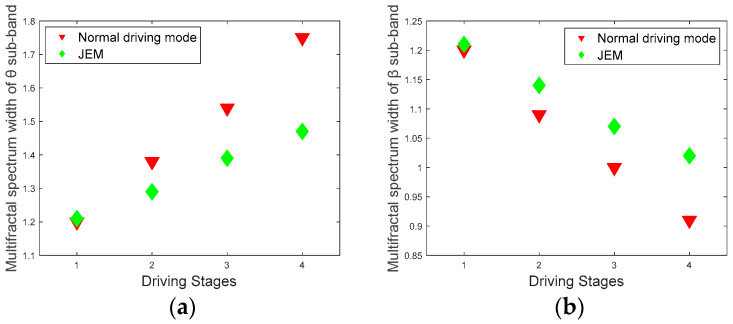
Multifractal spectrum width values. (**a**) multifractal spectrum width of the *θ* sub-band in two driving modes; (**b**) multifractal spectrum width of the *β* sub-band in two driving modes.

**Figure 13 brainsci-12-01199-f013:**
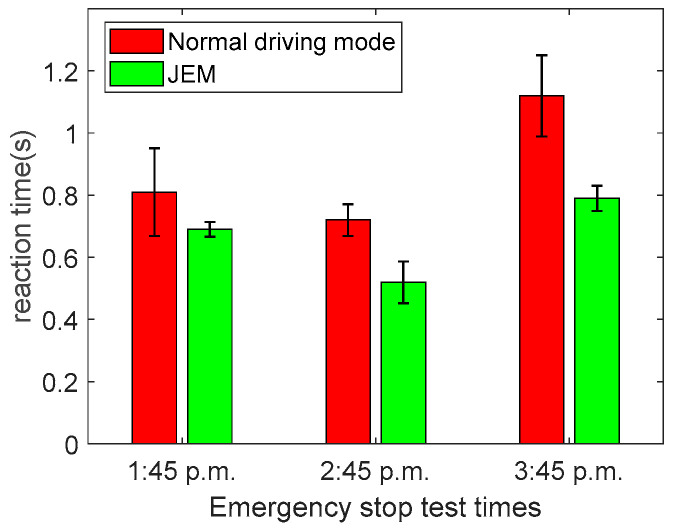
Emergency stop test.

**Table 1 brainsci-12-01199-t001:** The average value of the subjects’ wrong driving operations in the experiment.

Title 1	Normal Driving Mode	Musical Stimulation	JEM
Hit the vehicle	6.5	5.6	3.2
Hit the building	4.7	2.1	2.7
Driving deviation	5.6	5.2	4.8
Running a red light	8.2	9.6	7.5
Speeding	6.6	6.9	4.1
Gear mismatch	7.2	5.8	5.2
Incorrect use of turn signals	4.5	2.3	1.1
Abnormal flameout	11.3	7.2	3.6

## Data Availability

The effective data email: wangfuwangfeixue@163.com.

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
