# Peer review of "Study on the Effect of Judgment Excitation Mode to Relieve Driving Fatigue Based on MF-DFA"

_brainsci, 2022, doi:10.3390/brainsci12091199_

Round 1

Reviewer 1 Report

1. figures need to be enhanced, texts inside figures are not readable and the quality of figures is low.

2. the pros and cons of other methods that compared to this one are not very clear. for how many hours driving listening music works? what kind of music? are all  other mentioned methods were is the same driving scenario?

Reviewer 2 Report

Dear Authors,

First of all, thanks FOR YOUR WORK and effort for contributing with this interesting research about relieve fatigue on drivers. After a careful assessment of the paper, I believe the reviewed manuscript addresses a pertinent research problem (i.e. JEM for delay driving fatigue) for being considered as publishable in Brain Sciences. Of course, some points need to be addressed beforehand.

In general, the structure of the paper is adequate; however, the presentation can be clearer and more concise. In this regard, some concerns, queries and suggestions raised during my review must be addressed, in order to optimize the manuscript contents and its suitability for the journal:

·        ABSTRACT: Please, explain better the problem (driving fatigue)

·        INTRODUCCTION: Please, better describe the problem of fatigue and its effects on driving.

Devotes too much content to methods and too little about driving fatigue problem and how can reducing accidents, it would be convenient to compensate….

In the last paragraph of the introduction use for the first time the JEM concept, please explain here de JEM acronym, (In this study, we proposed JEM, which can improve the subjects' driving interest and increase the active cognitive load through the human-computer interaction of voice + but-  ton, and finally achieve the goal of delaying the occurrence of fatigue).

·        MATERIALS AND METHODS:  the sample is too small, is not equal (man/ woman).  We do not know their experience on driving. Please correct and increase the sample.

Please describe better the simulator and the driving scenarios

·  RESULTS: Study findings are adequately described, and Tables and graphics result helpful to support the reported along the text, but authors should consider better discussing the findings…as long as the sample increases.

·        DISCCUSSION: please make a discussion and expose (the answers and challenges that your research contributes to road safety and specifically to driving fatigue).

·        CONCLUSION: Adequate, and within the scope of the data. However, study limitations could be improved.

I think that the work and the design is good, but the results cannot be supported with such a sample.

Reviewer 3 Report

I enjoyed reading the paper. The authors have presented an interesting article about a Effect of Judgment Excitation Mode to Relieve Driving Fatigue Based on MF-DFA.

Abstract, overview
The abstract is a concise description of the work. The introduction is well structured, and it covers all the concepts investigated in the methodological part. The previous work is well presented and integrated. I consider that this work brings added value in the field and the specific objectives of the manuscript are well related to the previous work developed in this domain. 

Methodology
The research design used is appropriate in order to answer the research questions proposed by the authors. The methods are described properly. The results are clearly presented and are in relation to the concepts investigated.

Discussion and conclusions
The discussions are clear and concise. The conclusions are strongly related to the findings of the research work.

Format and style
All the format and style features were respected and are compliant with the requirements.

References
The format of the reference list fixes well to the specified format.

Plagiarism and any other ethical concerns about this study
I do not have any potential conflict of interest with regards to this paper.

Despite the good work done, there is still some room for improvement, as follows:

  • I think some more literatures should be added. Besides the mentioned HMI there are several other systems (like cost-effect BCI, eye-tracking) which are applied nowadays. It would be good to see the "effect of different web-based media" content on "human brain waves", as well as the additional applications of brainwave-based control like in examine the effect of different web-based media on human brain waves. It would improve the quality of the publication to mention the relationship between a cognitive psychological attention test and the attention levels determined by a BCI systems such as in an examination and comparison of the EEG based attention test with CPT and TOVA. In addition to BCI systems, mentioning other important human-computer interaction eye movement tracking would also improve quality, as such systems can be used in the analysis of programming technologies such as LINQ and algorithms, thus enabling, for example, cognition load or source code, algorithm description tools readability testing like in measuring cognition load using eye-tracking parameters based on algorithm description tools, in clean and dirty code comprehension by eye-tracking based evaluation using GP3 eye tracker and in analyse the readability of LINQ Code using an eye-tracking-based evaluation.

Round 2

Reviewer 2 Report

Thanks for your explications

Good work!

Author Response

Thank you for your suggestions on our articles!

Reviewer 3 Report

I accept the paper in the present form.

Author Response

(The authors gave the same response as above.)
